



# The impact of precipitation evaporation on the atmospheric aerosol distribution in EC-Earth v3.2.0

Marco de Bruine[1], Maarten Krol[1,2,3], Twan van Noije[4], Philippe Le Sager[4], and Thomas Röckmann[1]

[1]Institute for Marine and Atmospheric Research Utrecht, Utrecht University, Utrecht, The Netherlands
[2]Department of Meteorology and Air Quality, Wageningen University, Wageningen, The Netherlands
[3]Netherlands Institute for Space Research SRON, Utrecht, The Netherlands
[4]Royal Netherlands Meteorological Institute, De Bilt, The Netherlands

*Correspondence to:* M. de Bruine (m.debruine@uu.nl)

**Abstract.** The representation of aerosol-cloud interaction in global climate models (GCMs) remains a large source of uncertainty in climate projections. Due to its complexity, precipitation evaporation is either ignored or taken into account in a simplified manner in GCMs. This research explores various ways to treat aerosol resuspension and determines the possible impact of precipitation evaporation and subsequent aerosol resuspension on global aerosol burdens and distribution. The representation of wet deposition of aerosols by large-scale precipitation in the EC-Earth model has been improved by utilising additional precipitation related 3-D fields from the dynamical core IFS in the chemistry and aerosol module TM5. A simple approach of scaling aerosol release with evaporated precipitation fraction leads to an increase in the global aerosol burden (+7.8 to +15%, for different aerosol species). However, when taking into account the different sizes and evaporation rate of raindrops following Gong et al. (2006), the release of aerosols is strongly reduced, and the total aerosol burden decreases by -3.0 to -8.5%. Moreover, inclusion of cloud processing based on observations by Mitra et al. (1992) transforms scavenged small aerosol to coarse particles, which enhances removal by sedimentation and hence leads to a lower burden of small size aerosol by -10 to -11%. Finally, when these two effects are combined the global aerosol burden decreases by -11 to -19%. Compared to MODIS satellite observations, AOD is generally underestimated in most parts of the world in all model set-ups and although the representation is now physically more realistic, global AOD shows no large improvements in spatial patterns. Similarly, the agreement of the vertical profile with CALIOP satellite measurements does not improve significantly. However, aerosol resuspension after precipitation evaporation has a considerable impact on the modeled aerosol distribution and needs to be taken into account.

## 1 Introduction

Aerosols influence the energy balance of the Earth directly by interacting with solar and terrestrial radiation, and indirectly by impacting cloud formation. Even though the fundamental understanding of the interaction between aerosols and clouds has strongly improved over the past decade, translating and combining the wide range of contributing processes into parameterisations that can be applied in models on global scales introduces large uncertainties (Seinfeld et al., 2016) and dominates the uncertainty in climate projections (e.g. Fan et al., 2016; Boucher et al., 2013). Cloud microphysics involves processes of





different length and timescales. The large-scale hydrological cycle has to be linked to the condensation of water on individual aerosol particles at the small scales, the subsequent growth of particles, and the eventual rain-out of aerosol particles to the surface. To assess the effect of aerosols on our climate, models have been developed that calculate the distribution of aerosols based on emission and formation in the atmosphere, and include interaction with short- and longwave radiation (direct effect)

and with the hydrological cycle (indirect effect). Being very small in scale, the underlying microphysical processes cannot be modelled explicitly in GCMs and have to be represented by a set of parameterisations to remain manageable in terms of computational cost. Climate models widely vary in their representation of processes that they involve to model the aerosol burden (e.g. Textor et al., 2006). Differences found in model intercomparisons (e.g. AeroCom, http://aerocom.met.no) include varying source strengths, aerosol formation processes, numerical description of the aerosol size distribution, and removal processes.

This leads, among others, to large differences in the modelled vertical distribution (e.g. Kipling et al., 2016; Koffi et al., 2016) and SOA (Tsigaridis et al., 2014), showing that uncertainties in aerosol modelling are still paramount. The combination of the large (spatial and temporal) variability and the multitude of aerosol species, which are also difficult to measure at the relevant scales, makes it hard to validate the outcome of model estimates with in situ and satellite observations.

For most aerosol species, wet scavenging is the dominant removal process (e.g. Textor et al., 2006; van Noije et al., 2014).

Aerosols are incorporated in clouds and precipitation by nucleation and impaction scavenging (called in-cloud scavenging), and uptake by falling hydrometeors (called below-cloud scavenging). The ability of aerosols to act as cloud condensation nuclei (CCN) or the probability of coalescence upon collision with existing hydrometeors depends on aerosol size and hygroscopicity. Under certain conditions some aerosols might be incorporated in hydrometeors whereas others remain airborne (e.g. Rosenfeld and Mintz, 1988; Pruppacher and Klett, 1997). A substantial amount of the precipitation that forms aloft evaporates before

reaching the surface so the enclosed aerosols will be returned to the atmosphere. Ignoring this effect might thus introduce large errors in the simulated aerosol distribution. For example, uptake and resuspension of aerosol causes a redistribution of aerosol in the vertical column. Moreover, scavenged aerosols can dissolve and dissociate in the droplet water and be subject to aqueous phase chemistry. Upon subsequent evaporation of clouds or precipitation, the released aerosol distribution may be completely different from the initial scavenged aerosols (e.g. Wurzler et al., 2000). The release of aerosols from evaporated hydrometeors

in itself introduces additional uncertainties. As postulated by Gong et al. (2006), aerosols tend to remain incorporated in hydrometeors until complete evaporation of the host hydrometeor. Thus, besides the aerosol size distribution and chemical properties, information on hydrometeor distributions (i.e. cloud and raindrop) is necessary to correctly describe the relation between evaporated precipitation and the release of aerosol.

The representation of scavenging and release of aerosols in GCMs is often poorly documented. Most models adopt a straight-

forward approach and release aerosols proportional to the fraction of evaporated precipitation (e.g. ECHAM5-HAM (Stier et al., 2005), GOCART (Chin et al., 2000), SPRINTARS (Takemura et al., 2000)), but some models have a more advanced treatment. For example, AURAMS (Gong et al., 2006) takes the raindrop size distribution into account and only releases aerosol from evaporated droplets. EMAC, on the other hand, (GMXe SCAV module (Tost et al., 2006)) releases aerosols only upon complete evaporation of a cloud or falling precipitation.





In this paper we closely examine the removal and redistribution of aerosol by large-scale precipitation and re-evaporation. The primary aim is to include the effect of rain evaporation in the EC-Earth model and to verify if a more advanced treatment of evaporation will improve the comparison to observations. Starting from the existing implementation (van Noije et al., 2014), we gradually refine the numerical treatment of precipitation evaporation and analyse the impact on the simulated aerosol burdens. Here we consider simple treatments that merely act as an aerosol transport mechanism, and more advanced treatments in

which aerosols are processed by precipitation.

The outline of this paper is as follows. A description of the EC-Earth climate model and relevant modules and observational datasets is given in Sect. 2. A detailed description of the proposed new wet deposition scheme follows in Sect. 3. A thorough investigation of the impact of the changes to the wet deposition scheme and evaluation against MODIS and CALIOP datasets

is given in Sect. 4. This includes the changes in precipitation as well as the impact of different choices for resuspension and cloud processing. A discussion and general conclusions follow in Sect. 5.

## 2   Description of models and observations

The model used in this study is the global climate model EC-Earth version 3.2b (Hazeleger et al., 2010, 2012, http://www.ec-earth.org). This model consists of several Earth system components that can be coupled interactively. For this study, we have

employed the atmosphere-only configuration consisting of the general circulation model IFS, which includes the land-surface scheme H-TESSEL, and the chemistry and aerosol model TM5.

    A brief introduction of the dynamical core IFS is given in Sect. 2.1. The chemistry and aerosol module TM5 is described in Sect. 2.2, followed by a more detailed description of the representation of wet removal in Sect. 2.3. This includes an overview of the updates since the work by van Noije et al. (2014). A short explanation of the coupling and data exchange between the

IFS and TM5 modules in given in Sect. 2.4. Finally, in Sect. 2.5 we describe the satellite observations by MODIS and CALIOP that are used to evaluate the different model simulations and put the outcome of this modelling exercise into perspective.

### 2.1   IFS

The dynamical core of EC-Earth is based on cycle 36r4 of the Integrated Forecasting System (IFS) model (ECMWF, 2009) used by the European Centre for Medium-Range Weather Forecasts (ECMWF) for weather forecasting. The version of IFS used in

EC-Earth version 3.2b is modified for climate simulations and uses prescribed, climatological greenhouse gas concentrations, aerosol fields and land use following the forcing data of the WRCP Coupled Model Intercomparison Project Phase 5 (CMIP5). For example, direct radiative effects of aerosols are included based on the mass mixing ratios of various aerosol components calculated by the CAM model (Lamarque et al., 2012). Additionally, the model does not include indirect aerosol effects as a result of changing aerosol concentrations, because cloud formation is based on fixed CCN concentrations which only

differentiate between relatively clean air above oceans and more polluted air over land masses. For this study, we used IFS at



a spectral horizontal resolution of T255 (corresponding to 80 km) using 91 hybrid $\sigma$-pressure levels in the vertical and a time step of 45 minutes.

Another feature added to IFS is the option to apply nudging (e.g. Jeuken et al., 1996) for a number of prognostic atmospheric fields to reference values from reanalyses, in particular ERA-Interim (Dee et al., 2011). In the current set-up, vorticity and divergence of the wind field and atmospheric temperature are nudged throughout the whole atmosphere, whereas pressure
is nudged only at the surface. For the strength of this nudging a relaxation time constant of 6 hours is used. The nudging procedure takes place every timestep, however, the reference reanalysis data is imported on a 6-hourly time interval. For intermediate timesteps, reference values are calculated as a linear interpolation of preceding and subsequent data points.

## 2.2  TM5

Trace gases and aerosols in EC-Earth are simulated by the atmospheric chemistry and transport model TM5 (Krol et al., 2005;
Huijnen et al., 2010; van Noije et al., 2014), which is driven by the meteorology calculated in IFS. Currently, the simulated chemistry does not feed back to IFS, but this two-way coupling between aerosols and meteorology is under development. TM5 simulates the evolution of different reactive and non-reactive gases using a modified version of the CB05 scheme (Yarwood et al., 2005; Williams et al., 2017). A detailed overview of used emissions can be found in van Noije et al. (2014). Aerosol microphysics is described using the modal scheme M7 (Vignati et al., 2004) with an additional bulk description of semi-
volatile species by the chemical equilibrium model EQSAM (Metzger et al., 2002). The TM5 model was recently updated to the massively parallel version TM5-mp (Williams et al., 2017) to run more efficiently on computers geared towards high-performance parallel computing. Simulations of chemistry and aerosols in this research are done on a resolution of 2° latitude by 3° longitude. In the vertical, TM5 uses 34 vertical hybrid $\sigma$-pressure levels, which are a subset of the 91 levels used in IFS. The performance of this model is evaluated and documented in intercomparison projects, e.g. AeroCom. Compared to
observations, the AOD field simulated in TM5 has generally been too low (see e.g. Aan de Brugh et al. (2011); van Noije et al. (2014)). This has been a point of attention for several cycles of the model and although substantial improvements have been made, the underestimation has not been completely resolved as of yet.

## 2.3  Aerosols and wet deposition

Aerosols in TM5 are described by the size-resolved modal microphysics scheme M7 (Vignati et al., 2004). This scheme
uses seven lognormal size distributions representing the five species sulphate ($SO_4$), particulate organic matter (POM), black carbon (BC), sea salt (SS) and mineral dust (DU). These are distributed over four water-soluble size modes (nucleation, Aitken, accumulation and coarse) and three insoluble modes (Aitken, accumulation and coarse). Each of these modes contains a sub-set of the aerosol species under the assumption of a complete internal mixture within each mode. As such, the mass of the included species are tracked by separate prognostic variables, but there is only one prognostic variable for the total number of
aerosols in each mode. Together with the mass and number, each mode has a fixed width with which the size distribution of the aerosols can be described. A more detailed overview of the implementation of M7 in TM5 is given by Aan de Brugh et al. (2011) and van Noije et al. (2014). The semi-volatile species treated by EQSAM include the aerosols of ammonium nitrate,





**Table 1.** In-cloud scavenging efficiency factors used in the wet deposition scheme. Values adopted from Bourgeois and Bey (2011).

|  | Soluble | | | | Insoluble | | |
| --- | --- | --- | --- | --- | --- | --- | --- |
|  | Nucleation | Aitken | Accumulation | Coarse | Aitken | Accumulation | Coarse |
| $T > 0°C$ | 0.06 | 0.25 | 0.85 | 0.99 | 0.2 | 0.4 | 0.4 |
| $-35°C < T < 0°C$ | 0.06 | 0.06 | 0.06 | 0.75 | 0.06 | 0.06 | 0.4 |
| $T < -35°C$ | 0.06 | 0.06 | 0.06 | 0.06 | 0.06 | 0.06 | 0.06 |

but describes mass only. For the calculations of e.g. optical properties of aerosols this mass is assumed to be condensed on the soluble accumulation mode particles of M7.

Due to the coarse resolution of global climate models, clouds and precipitation cannot be completely resolved. This is reflected in a distinction between large-scale (stratiform) and convective precipitation and clouds. The large-scale precipitation is described using variables like cloud cover and water content. Convective precipitation is considered sub-grid and is therefore parameterised. Following this distinction, the analysis in this study focuses on large-scale precipitation. This means that when precipitation is discussed, this does not include convective precipitation, unless stated explicitly.

Removal of aerosols by large-scale precipitation is simulated in the TM5 model by prescribed efficiency factors for the different M7 aerosol modes and bulk aerosol types. These factors are defined as the scavenging efficiency compared to the scavenging of the completely soluble species $HNO_3$ in the cloudy part of a gridbox, which is calculated by the description of Roelofs and Lelieveld (1995). A more detailed description of the scavenging by stratiform precipitation can be found in van Noije et al. (2014), although several changes have been made to the scheme since then. These changes are documented in this paper. In the current version, a distinction is made for in-cloud scavenging between liquid, mixed and ice stratiform clouds (Stier et al., 2005), depending on the local temperature (Croft et al., 2010). Boundaries are $0°C$ and $-35°C$ as shown in Table 1, to include the differences in behaviour of aerosols to act as cloud and/or ice condensation nuclei. Revised values of scavenging efficiencies are provided by Bourgeois and Bey (2011). For bulk aerosols the in-cloud scavenging efficiency is set equal to the scavenging efficiency of soluble accumulation size aerosols, as their mass is assumed to be condensed on this M7 mode.

The approach for below-cloud scavenging does not differentiate between soluble and insoluble aerosol modes (Stier et al., 2005), but between aerosol mass and number as shown in Table 2. Values are updated based on the work by Croft et al. (2009) and are substantially lower than the previously used values based on Dana et al. (1975). The bulk aerosols described by EQSAM use a fixed below-cloud scavenging efficiency of 0.004.

The combination of coarse resolution and meteorology that updates only every 6 hours can lead to a strong overestimation of scavenging in case of incomplete coverage of grid cells by large-scale clouds and precipitation. During the calculation of wet deposition, a cloud-free and cloudy part of a gridbox are defined, with scavenging only affecting tracer concentrations in the latter. However, this distinction between cloudy and clear part is not carried to the next timestep, artificially introducing (subgrid) mixing. Consequently, scavenging will be (partly) applied to tracers that are situated in the clear part of the gridbox.





**Table 2.** Below-cloud scavenging efficiency factors used in the wet deposition scheme. Values adapted from Croft et al. (2009).

|        | Nucleation | Aitken | Accumulation | Coarse |
|--------|------------|--------|--------------|--------|
| Number | 0.02 | $1 \times 10^{-3}$ | $3 \times 10^{-4}$ | 0.3 |
| Mass   | $2 \times 10^{-3}$ | $2 \times 10^{-4}$ | 0.03 | 0.7 |

To correct for this, an exponentially decreasing factor is introduced to scale down the scavenging efficiencies, simulating a mixing timescale of 6 hours when using a 3 by 2 degree resolution.

## 2.4 Coupling of meteorological fields

Data exchange within the EC-Earth framework takes place through OASIS3-MCT coupler (Valcke, 2013; Craig et al., 2017) on a 6 hourly basis. In the setup used for this research, a one-way coupling is used between IFS and TM5. Meteorological fields of IFS are used to drive the calculations in TM5, but there is no feedback of the aerosols and chemistry in TM5 to the processes calculated by IFS. A detailed overview of the coupling between IFS and TM5 is given in van Noije et al. (2014), in their Sect. 2.2.2. In the standard configuration, no structural changes are made to the way meteorology is used in the standalone

TM5 model, which relied on operational forecasts of meteorological variables or reanalysis datasets like ERA-Interim to drive transport, removal and other meteorological dependant processes for chemical species. For example, the current wet deposition parameterisation in TM5 (hereafter called BASE) does not make use of 3-D precipitation fields available in IFS, but recalculates precipitation (formation) based on liquid and ice water content. In this procedure, precipitation evaporation is ignored and all precipitation formed within a cloud is assumed to reach the surface. To still be consistent with the IFS precipitation, the

complete vertical precipitation profile is scaled to match the IFS value for surface large-scale precipitation. As will be shown in Sect. 4.1, this approach underestimates precipitation (Fig. 7) and consequently in- and below-cloud scavenging. In contrast, ignoring precipitation evaporation prohibits resuspension of scavenged aerosol. These two effects have an opposite sign and one aim of this study is to assess the net effect of the changed precipitation and inclusion of evaporation.

In this work, the coupling is extended with three extra 3-D precipitation fields: falling precipitation, precipitation formation,

and precipitation evaporation. Fig. 1 visualises the vertical regridding of these quantities from the resolution used in IFS to the ones used in TM5. In IFS, falling precipitation is defined as the mass flux of precipitation leaving the bottom of a gridbox (kg m$^{-2}$ s$^{-1}$). Therefore, we sample the values of this quantity at the pressure levels shared by both modules. Precipitation formation and evaporation are defined as the average mass rate of change per volume (kg m$^{-3}$ s$^{-1}$) in a gridbox. To be consistent with falling precipitation, the value of formation and evaporation sent from IFS to TM5 is the volume-weighted

average of the formation and evaporation in the gridboxes above a shared pressure level. The additional precipitation fields are sent as a time average of the preceding 6 hours before data exchange.





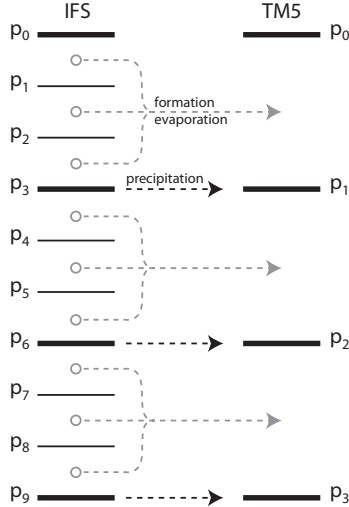

**Figure 1.** Regridding of additional meteorological fields in the coupling process between IFS and TM5. The mass flux of falling precipitation is directly sampled at the pressure levels shared by both modules. For precipitation formation and evaporation a volume-weighted average is calculated for multiple IFS pressure levels corresponding to one TM5 pressure level.

## 2.5 Observational data

To validate the results and to put the differences between the simulations into perspective, model output is compared to MODIS and CALIOP measurements. From the MODIS Level 3 Atmospheric Gridded Product (Platnick et al., 2015) the monthly-mean aerosol optical depth is used to evaluate the performance of the model on a global scale. An average is constructed from the combined Dark Target and Deep Blue retrievals of both Terra and Aqua satellites. However, the MODIS AOD data are total column values and because the adjustments made to the model directly influence the vertical distribution of aerosols, an

additional comparison is made to CALIOP observations. We adopt the framework used in Koffi et al. (2012, 2016) and compare aerosol extinction from the different simulations with the CALIOP aerosol extinction profile data provided as benchmark data within the AeroCom project (Koffi et al., 2012; http://aerocom.met.no/databenchmarks.html). As stated in their work only night-time CALIOP overpasses are used as these yield more reliable results. Since CALIOP coverage is adequate to evaluate the mean aerosol climatology on seasonal time scales and over sub-continental areas, we do not collocate model output with

CALIOP data. Also similar to Koffi et al. (2012, 2016) the simulation period (Jan 2005 to February 2006) is not covered by the CALIOP measurements. Instead, we use the multi-year (2007-2009) average as a climatology to evaluate our model output.





## 3 Implementation of aerosol re-evaporation

### 3.1 New wet deposition scheme

By incorporating TM5 into the EC-Earth framework we unlock a large source of additional meteorological information. Limited availability of meteorological fields in forecasts or reanalysis datasets do no longer limit calculations related to the chemistry of gases and aerosols. Using data calculated in IFS, we can implement precipitation evaporation and subsequent tracer resuspension in the TM5 model. To do so, we introduce diagnostic variables for each tracer in falling precipitation. Scavenging and evaporation of gases and aerosol are calculated for each column of the grid following the precipitation, i.e. from the top of the atmosphere towards the surface. Formed precipitation is assumed to either reach the surface or evaporate within the same model timestep and scavenged aerosol mass is either deposited to the ground or resuspended. For each model level in an atmospheric column we have the following balance within a timestep:

$$M_{i,k} = M_{i,k-1} + S_{i,k} - E_{i,k} \tag{1}$$

where $M_{i,k}$ is the amount of tracer $i$ contained in precipitation at level $k$, where $k$ counts from the top down. $M_{i,k-1}$ is the amount of tracer at level $k-1$ directly above the current level $k$ and transported into the current level by falling precipitation. $S_{i,k}$ is the total mass of tracer $i$ scavenged (in- and below cloud) at level $k$ and $E_{i,k}$ the total mass of tracer $i$ re-evaporated at level $k$.

When cloud processing is ignored, scavenged species are assumed to be homogeneously distributed in the hydrometeors, so the fraction of evaporated tracer is equal to the fraction of evaporated precipitation. Optional extensions to this approach are described in Sect. 3.2. When precipitation formation and evaporation take place within the same gridbox, we assume these processes are separated and happen in different parts of the gridbox, i.e. the mass available for evaporation at level $k$ is the mass transported downward from level $k-1$ . This gives:

$$E_{i,k} = f_k M_{i,k-1} \tag{2}$$

with $f_k$ representing the evaporated precipitation fraction. Combining these equations and eliminating $E_{i,k}$ yields

$$M_{i,k} = (1 - f_k) M_{i,k-1} + S_{i,k} \tag{3}$$

### 3.2 Choices on processing of aerosol species

Re-evaporation of aerosols introduces the necessity to take into account additional details regarding the interaction between aerosols and precipitation. This interaction involves multiple sets of size distributions (aerosols and precipitation) of a vast amount of particles, spanning multiple orders of magnitude in mass and number. As described above, the representation of the aerosol distribution in TM5/M7 is reduced to mass and number using lognormal size distributions with fixed widths. For precipitation a bulk approach is used and only information on precipitating mass is available. Processes determining the uptake of an aerosol in hydrometeors are accounted for in the scavenging coefficients described above. However, these do not





provide information on the fate of the aerosol once incorporated in a hydrometeor. Without a detailed description of the aerosol content inside the hydrometeors it is hard to quantify how aerosols behave inside precipitation. Nevertheless, we can explore the influence of precipitation re-evaporation using two sets of assumptions: aerosol mass resuspension and aerosol number processing. Below, we outline the details and possibilities for both assumptions.

### 3.2.1 Mass resuspension

Evaporation of gases directly follows from the fraction of evaporated precipitation, justified by the fact that mass transfer mainly occurs via diffusion. One could apply this to the re-evaporation of aerosols as well, but aerosol material tends to remain in the host hydrometeor and only returns to the aerosol phase once the hydrometeor completely evaporates (Gong et al., 2006, hereafter G06). This mechanism breaks the simple proportionality between precipitation evaporation and aerosol resuspension. When not all precipitation evaporates, only the smaller droplets disappear and release their aerosol mass. While the larger droplets contribute relatively more to the total evaporated rainwater, because evaporation is proportional to the droplet surface, their suspended aerosols are not released. This reasoning does not hold for solid precipitation, but our model will treat all precipitation similarly. Therefore, in the remainder, where water or rainwater are mentioned this includes both liquid and solid precipitation. Although ice microphysics is substantially different from warm-phase microphysics, these processes are too poorly understood and would introduce additional uncertainties unnecessarily complicating our analysis.

G06 proposed a relation to account for these effects and combined them in a correction factor $\epsilon'$ (first introduced by Barth et al., 1992) to be multiplied with the evaporated precipitation fraction $f$ in Eq. 3. This relation is based on the Marshall-Palmer (MP) raindrop size distribution (Marshall and Palmer, 1948), which is also underlying the below-cloud scavenging efficiencies presented in Table 2. Combining equations (12) through (14) of G06 yields an expression of $\epsilon'$ that depends only on $f$:

$$\epsilon' = \left[1 - \exp(-2\sqrt{f})\left(1 + 2f^{\frac{1}{2}} + 2f + \frac{4}{3}f^{\frac{3}{2}}\right)\right](1-f) + f^2 \tag{4}$$

This relation is plotted in Fig. 2. Following the rationale that aerosol is only released if a hydrometeor completely evaporates, the fraction of released aerosol is lower than the fraction of evaporated precipitation. The discrepancy is largest when about 57% of the precipitation water evaporates, which releases only 20% of the aerosol mass into the atmosphere.

### 3.2.2 Number processing

Every hydrometeor starts out as an aerosol particle with water condensing on its surface. However, looking at an average drop, this origin can barely be recognised. After initial activation, droplets continue growing and will collide and combine with other droplets and aerosols. Because of this process, called collision-coalescence, one cloud or raindrop contains a multitude of aerosol particles. However, observations (e.g. Mitra et al., 1992, hereafter M92), show that generally each evaporated raindrop leaves behind only one aerosol particle. This means that while aerosols are suspended in the water of a droplet, processes take place causing the particles to clump together. The intricate details of dissolution, dissociation and complex aqueous chemistry are outside the scope of this work and we will only explore the two extremes of the processing spectrum: no interaction or



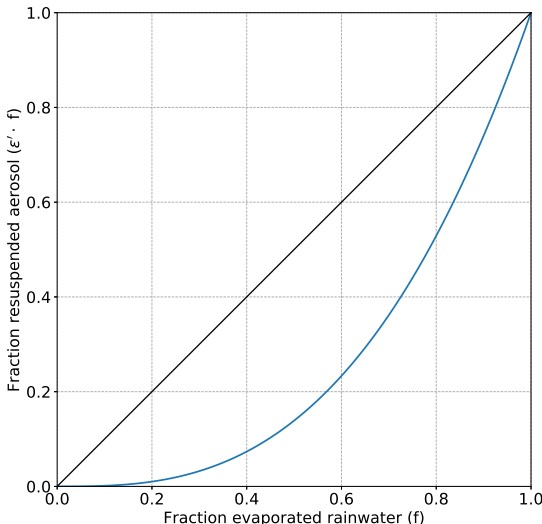

**Figure 2.** Relation between fraction of evaporated rainwater $f$ and the fraction resuspended aerosol $\epsilon' \cdot f$.

complete clumping of all aerosol mass inside a droplet.

When number processing is neglected, all aerosol tracers (both mass and number) are treated individually so that their characteristics are retained. Besides the optional adjustment of resuspended mass fraction (Sect. 3.2.1), the treatment of aerosols is similar to gaseous species and all tracers have individual diagnostic variables to represent their suspension in rainwater. In this case, the number of particles contained in precipitation for each mode is described by an expression analogous to Eq. 3.

When accounting for number processing, aerosol tracers in the model do no longer have individual diagnostic variables to represent suspension in rainwater. Instead, the mass of one species (e.g. sulphate) is aggregated in one diagnostic tracer, while aerosol number for this species is not tracked at all. Instead, the raindrop size distribution (RSD) is used to determine the number of re-evaporated aerosols, i.e. the number of released aerosols will be equal to the amount of evaporated raindrops. To

be consistent with other parts of the model, we again assume the Marshall-Palmer RSD. To calculate the number of evaporated raindrops (and hence the released aerosol particles) from the RSD, it is assumed that the evaporated mass scales with the total surface of a droplet. With this assumption, all droplets contribute to the evaporation of water, but only the smaller droplets completely evaporate and disappear. This leads to the following expression for the number of evaporated aerosols $N_k$ at level $k$:

$$N_k = \epsilon' f_k \frac{N_0}{\lambda} \tag{5}$$





where $\epsilon'$ and $f$ have the same meaning as in previous equations. $\lambda$ (mm$^{-1}$) is a slope parameter in the MP RSD and $N_0$ (m$^{-3}$ mm$^{-1}$) is the number concentration of droplets per unit radius ($r$) in the Marshall-Palmer RSD in the limit $r \to 0$. A detailed derivation of this relation is given in appendix A.

Released aerosol is assumed to follow a lognormal size distribution and transfered to the soluble accumulation and coarse modes, based on the evaporated aerosol mass and number. All aerosols with a dry diameter smaller than 1 $\mu m$ are considered accumulation size aerosol, while larger aerosols are returned to the coarse mode. This threshold is consistent with the aerosol modes in M7. The fraction ($F$) of a lognormal distribution below a certain threshold $D_c$ is given by:

$$F = \frac{1}{2} \mathrm{erfc} \left( -\frac{\ln(D_c/\widetilde{D})}{\sqrt{2}\ln(\sigma)} \right) \tag{6}$$

where $\widetilde{D}$ is the median diameter and $\sigma$ the geometric standard deviation. This equation can be applied to calculate the fraction of both mass and number of the released aerosol below a given threshold when $\widetilde{D}$ is replaced by the number median ($D_n$) and mass median diameter ($D_m$) respectively. These are in turn calculated as

$$D_n = \left( \frac{6E_k}{\pi N_k \rho} \right)^{1/3} \exp \left( -\frac{3\ln^2(\sigma)}{2} \right) \tag{7}$$

$$D_m = D_n \exp(3\ln^2(\sigma)) \tag{8}$$

Because the representation of aerosol in M7 assumes internally mixed aerosol populations, only one population of aerosols can be re-suspended, i.e. $E_k$ in Eq. 7 is the sum of released mass of all aerosol species $E_{i,k}$, and the aerosol density $\rho$ is calculated as the volume weighted average of the densities of the released aerosol masses.

$$E_k = \sum_i E_{i,k} \tag{9}$$

$$\rho = \frac{\sum\limits_i E_{i,k}}{\sum\limits_i E_{i,k}/\rho_i} \tag{10}$$

### 3.2.3 Overview

In the new wet deposition scheme of the coupled EC-Earth-TM5 system, precipitation can thus act as a transport mechanism as well as a processing medium for aerosols. When the model is set up to ignore number processing, the sequence of repeated scavenging, transport and evaporation is calculated for each number and mass tracer individually. When the release of single aerosol particles is based on the Marshall-Palmer RSD as described in Sec. 3.2.2, it is no longer necessary to track each tracer. This representation is displayed in Fig. 3. Scavenged aerosol mass in different modes of the same species (e.g. sulphate) is added together, leaving five diagnostic tracers for all aerosol species represented in the M7 scheme. The number of aerosol

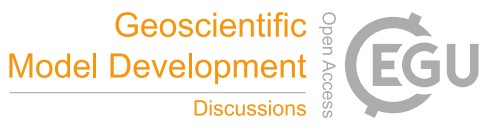

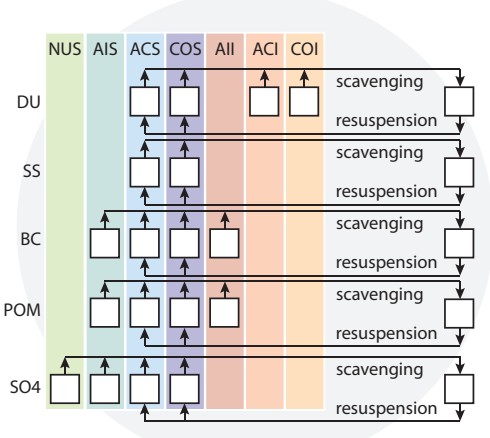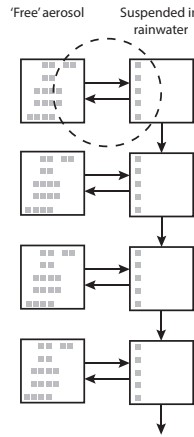

**Figure 3.** Left: Close-up of the transfer between 18 mass tracers in M7 and suspended aerosol tracers. Abbreviations stand for M7 modes. The first two letters refer to mode size and the last to mode solubility, e.g. ACS is the soluble accumulation mode. Note that bulk aerosol species are not divided over multiple tracers and therefor do not need aggregation in the deposition scheme. Right: Overview of the new wet deposition scheme when number processing is applied. The atmospheric 'free aerosol' tracers of M7 and the new wet diagnostic deposition scheme with 5 aggregated tracers (grey dots) are transferred by scavenging and resuspension (arrows) on each level.

tracers suspended in precipitation are no longer tracked, but the number of resuspended particles is based on the number of evaporated hydrometeors (Eq. 5). Resuspended mass is calculated with Eq. 3 applying the correction factor from Eq. 4 and distributed over the accumulation and coarse modes of the corresponding tracers. The distribution of the mass over these two
10  modes is determined by Eq. 6. The exchange between the M7 mass tracers and diagnostic suspended aerosol mass variables is calculated column-wise and top-down as visualised in the right-hand panel of Fig. 3.

### 3.3 Simulations

To disentangle different effects of the introduced changes to the wet deposition scheme the following series of model simulations is performed. To establish a benchmark, the first model simulation (BASE) uses the status quo of the wet deposition
15  scheme described above in Sect. 2.3, i.e. internal recalculation of precipitation formation in TM5 and no evaporation of precipitation and therefore no resuspension of aerosol. The other simulations use the updated meteorology and have different combinations of choices for mass resuspension and number processing. Fig. 4 shows a grid based on the options for the representation of aerosol release, having mass resuspension on one axis and number processing on the other. The different simulations are placed on this grid to visualise the relation between them. The SMPL simulation provides the simplest implementation of aerosol re-evaporation: mass resuspension follows the evaporated water fraction and does not take into account
number processing. The TRSP simulation adds to this by using the correction factor $\epsilon'$ (Eq. 4) from G06 for mass resuspension instead. Both simulations do not change properties of aerosols in precipitation. Therefore, in these two simulations precipita-





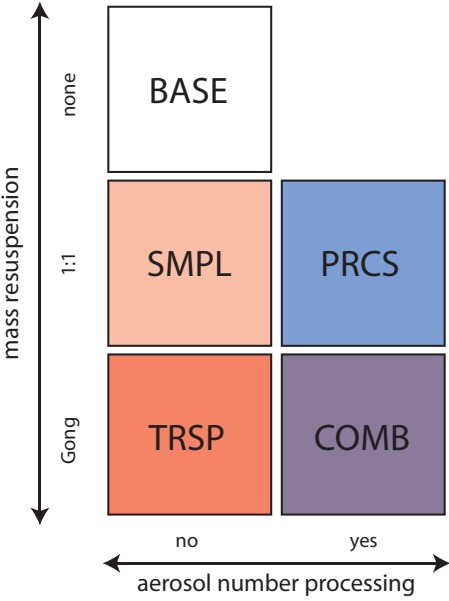

**Figure 4.** Overview of the simulations in this work. The position of the individual simulations indicates used choices for mass resuspension and number processing of aerosols. Please note that the BASE simulation uses different meteorology as stated in the text.

tion acts as an aerosol transport mechanism, vertically redistributing their number and mass. In the PRCS simulation number processing is applied and evaporated hydrometeors release a single aerosol. The released aerosol number is calculated from the raindrop number concentration (Eq. 5). For mass resuspension the simple 1:1 approach is used. The COMB simulation combines number processing with the mass resuspension relation of G06. All simulations consider the period Jan 2005 - Feb 2006, of which the first two months of the simulations are considered spin-up and disregarded in the analysis. The dynamics of IFS are nudged to the ERA-Interim database to ensure realistic meteorology, as described in Sect. 2.1.

## 4   Results

With the new implementation TM5 no longer recalculates precipitation, but receives the 3-D precipitation field directly from IFS. Also, with the introduction of evaporation there is no longer a need to rescale precipitation to surface values. This brings substantial changes to the vertical precipitation profile, which will be discussed in Sect. 4.1. Consequences of the new representation of precipitation will be shown in Sect. 4.2. In Sect. 4.2.1 we will investigate subsequent increased removal aloft and release of aerosol at lower levels and the resulting changes to the atmospheric aerosol burdens, using the SMPL simulation. The effect of the Gong relation for mass resuspension in the TRSP simulation will be discussed in Sect. 4.2.2 and Sect. 4.2.3 will describe number processing in the PRCS and COMB simulations. Finally, we will present a comparison of the model outcome to MODIS satellite observations of column integrated AOD and vertical extinction profiles measured by the CALIOP satellite in Sect. 4.3.





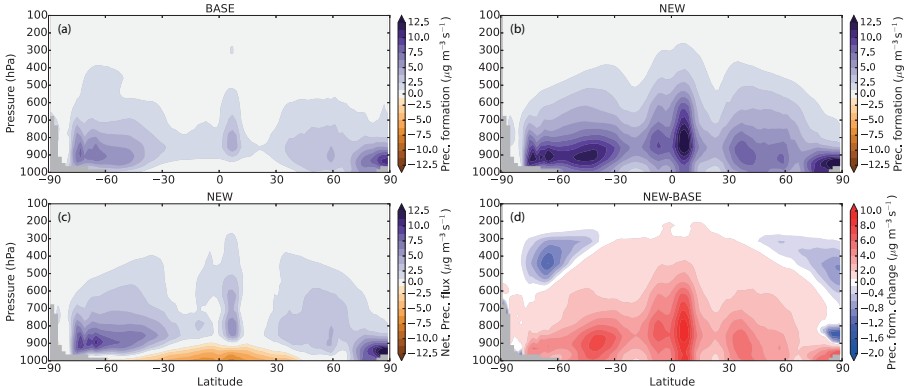

**Figure 5.** Zonal mean annual mean precipitation formation rate ($\mu$g m$^{-3}$ s$^{-1}$) in a) the BASE simulation and b) other simulations and d) the differences between these. c) Net precipitation flux (formation − evaporation) in other simulations.

## 4.1 Impact on precipitation

Before analysing the aerosol distribution, we first investigate how large-scale precipitation (LSP) changes between the patterns recalculated in the BASE simulation and the directly imported values from IFS used in all other simulations. Zonal mean precipitation formation rates for the BASE and new scheme are shown in Fig. 5, together with the difference between them. Net precipitation fluxes (formation − evaporation) in both schemes are relatively similar (panel a and c), with high values at the North Pole, Southern Hemisphere stormtracks and tropics. However, only the new scheme includes evaporation. Hence, the actual formation rates in the new scheme are much higher (panel b). Especially in the tropics, where evaporation is high, this leads to large differences in precipitation formation (panel d). Globally averaged, precipitation formation increased to 2.5 times the amount calculated in the BASE simulation.

Fig. 6 shows vertically integrated evaporation as a fraction of local precipitation formation, showing a strong latitudinal dependence. On the global scale, 60% of the formed precipitation evaporates before reaching the surface in the new scheme. In the subtropics, this value is close to and sometimes even exceeding 100%. Note that precipitation in IFS is treated prognostically, so that due to horizontal transport of precipitation, evaporation can exceed precipitation formation locally. From this maximum in the subtropics, the evaporation fraction gradually decreases to values below 20% over the poles. There, clouds and precipitation have a more stratiform character and are situated closer to the surface which reduces the amount of evaporation. Also visible is an increase in evaporation over the Northern Hemisphere in summer. This effect is less visible is the Southern Hemisphere, because of the smaller fraction of land mass.

A schematic sketch of the vertical profile of precipitation, precipitation formation and evaporation is given in Fig. 7 to clarify the consequences of ignoring evaporation in the wet deposition scheme of the BASE simulation. Precipitation rates from IFS (blue line) and initial recalculated values in TM5 (dotted red line) do not differ substantially. But by ignoring evaporation, the recalculation would remove too much water from the atmosphere. To prevent this, the complete vertical profile is rescaled to





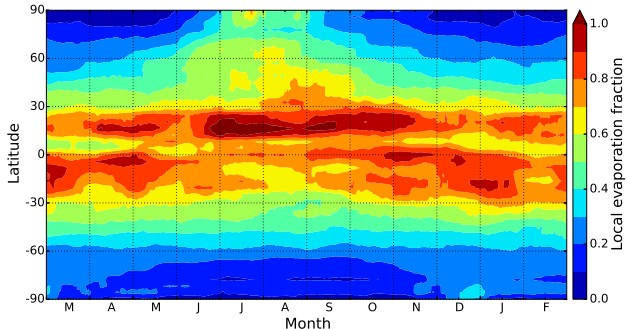

**Figure 6.** Column total, zonally averaged evaporation expressed as a fraction of local precipitation formation for March 2005 to February 2006. Values shown are smoothed by a 30-day running mean.

values of precipitation reaching the surface to be consistent with the amount of precipitation reaching the surface in IFS. But this procedure leads to an underestimation of both precipitation formation and precipitation throughout the vertical column. The largest difference is found at the cloud base and from there it reduces to zero at the surface where recalculated and imported precipitation are the same by definition.

An exception to this general underestimation is found over the poles, where precipitation formation is overestimated. This can be linked to the combination of widespread, low altitude stratiform precipitation and high ice clouds. In the BASE scheme all overhead clouds are assumed to form precipitation when non-zero precipitation is found at the surface. However, in reality (and in the new scheme) the high ice clouds form little or no precipitation which also quickly evaporates before reaching the surface.

## 4.2 Impact on aerosol burdens

Regardless of the choices made for evaporation or processing, the increased precipitation formation and subsequent evaporation will introduce a downward flux of aerosol within the atmosphere. The resulting aerosol burden, however, is not only determined by precipitation. Processes like advection, convection and sedimentation will mitigate or enhance the initial offset. For instance, the aerosol released by precipitation evaporation is redistributed by advection and convection, and cloud processing may release larger aerosol particles more susceptible to sedimentation. The TM5 model allows for a detailed analysis because all processes affecting the aerosol distribution are budgeted. The global burdens of the aerosol species in all simulations are displayed in Table 3, with zonal mean patterns shown in Fig. 8. In the following paragraphs we will analyse the effect of the different representations of aerosol release on the distribution of these aerosol species.

A more in-depth description of the changes in (removal) processes and a complete overview of their global impact on aerosol burden are given in Appendix B. All numbers mentioned in the main text are summarised in Table B1 or can be deduced from the values stated there.





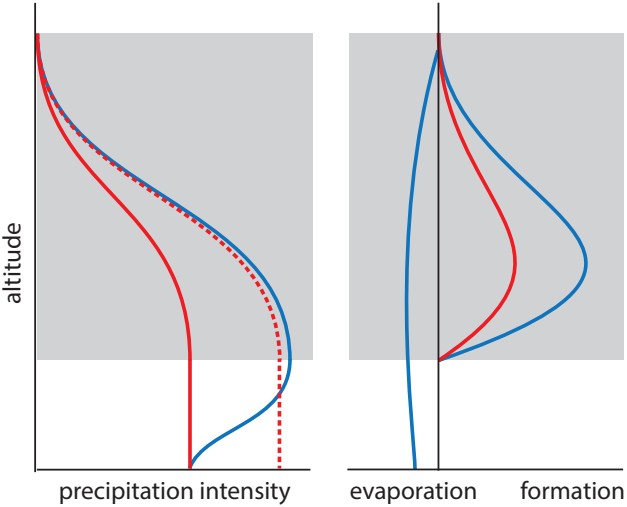

**Figure 7.** Schematic comparison of left) precipitation and right) precipitation formation (positive) and evaporation (negative). BASE recalculated (dashed) and rescaled (solid) profiles are shown in red, IFS values used in all other simulations in blue. The shaded area denotes the vertical cloud extent.

**Table 3.** Annual average global atmospheric burdens (Tg) of the individual aerosol species and global average AOD. Relative change with respect to the BASE simulation denoted between parentheses.

|  | BASE | SMPL | | TRSP | | PRCS | | COMB | |
|---:|---|---|---|---|---|---|---|---|---|
| Sulphate | 2.29 | 2.64 | (+15%) | 2.17 | (-5.3%) | 2.04 | (-11%) | 1.92 | (-16%) |
| POM | 2.29 | 2.48 | (+8.3%) | 2.11 | (-7.9%) | 2.03 | (-11%) | 1.85 | (-19%) |
| Black carbon | 0.202 | 0.218 | (+8.0%) | 0.184 | (-8.5%) | 0.180 | (-10%) | 0.163 | (-19%) |
| Mineral dust | 9.27 | 9.99 | (+7.8%) | 8.99 | (-3.0%) | 9.50 | (+2.5%) | 8.26 | (-11%) |
| Sea salt | 5.13 | 5.89 | (+15%) | 4.81 | (-6.3%) | 5.10 | (-0.57%) | 4.54 | (-11%) |
| Nitrate | 0.082 | 0.081 | (-1.1%) | 0.080 | (-2.7%) | 0.105 | (+27%) | 0.089 | (+7.9%) |
| AOD | 0.105 | 0.122 | (+16%) | 0.099 | (-5.9%) | 0.088 | (-16%) | 0.082 | (-22%) |

### 4.2.1 Precipitation as a transport mechanism

In the SMPL simulation, evaporation is introduced without adding aerosol processing inside precipitation and without taking into account the inhomogeneous evaporation of different raindrops according to the G06 relation. By comparing BASE and SMPL simulations we isolate the effect of transport induced by uptake and release of aerosols by precipitation. Compared to the BASE simulation, SMPL shows an 15% increase of the pure soluble species sulphate and sea salt, while the other species (POM, BC and dust) show an increase of about 8%. Nitrate shows a slight decrease of -1.1%.





The increase in precipitation formation leads to aerosol scavenging almost twice as strong for all species in the SMPL compared to the BASE simulation. However, the introduced evaporative release balances or even slightly exceeds the initial increase of scavenging, which results in generally higher aerosol burdens. Scavenging is a first order loss process and limits it's own potential to remove aerosols when local concentrations are not fully replenished by other processes. With the relatively high scavenging efficiencies this process is strongest for pure soluble species and hence leads to the large response of sulphate

and sea salt. Because scavenging is such a strong process this occurs for all areas under the influence of wet deposition.

Although less aerosols are removed by wet deposition over all, scavenging and resuspension of aerosol do not occur at the same altitude. As a net result aerosols are brought closer to the surface, i.e. the introduction of evaporation leads to a downward transport. Once released from precipitation, aerosols can immediately be re-scavenged, but can also be transported away by advection and convection and remain in the atmosphere. This mechanism leads to the pattern visible in the second column of Fig. 8: an increase near the surface and a decrease aloft. The decreases aloft are relatively small because the initial

removal by scavenging is counteracted by transport carrying excess aerosol from below. Additionally, at the highest altitudes the aerosol burdens again show an increase. This can be linked to the treatment of high (ice) clouds in the BASE simulation as explained in Sect. 4.1. The erroneous precipitation from high ice clouds calculated in the BASE simulation (see Fig. 5d) leads to a very long fetch for below-cloud scavenging and an overestimation in aerosol removal. In the SMPL (and all other) simulation precipitation formation from these clouds (Fig. 7) evaporates quickly and removes substantially less aerosol. At the

high latitudes this pattern is confined closer to the surface and the influence of ice clouds starts at lower altitudes.

Sea salt shows a pattern different from the other aerosol species, with a strong decrease in aerosol burden near the surface at the poles. This decrease is connected to the nature of precipitation at these latitudes in combination with the relatively large size of sea salt aerosol. Precipitation occurs relatively close to the surface and is predominantly stratiform. Because of this, precipitation has short falling distances and evaporation is low (Fig. 6). Also, convection here is weak and cannot carry the

large sea salt aerosol upward and away from the regions of precipitation.

Similar to the M7 aerosol species, wet deposition becomes less efficient in removing nitrate aerosol. However, chemical production of nitrate aerosol (in the form of ammonium nitrate) only occurs when the concentration of ammonia is sufficiently high to not be completely neutralised by sulphate. The increased burden of sulphate further impedes the formation of ammonium nitrate and results in a slight decrease in nitrate aerosol.

### 20  4.2.2   Change in mass resuspension

As put forward in G06, assuming a simple proportionality between evaporation of precipitation and resuspension of aerosol overestimates the release of aerosol. To quantify the effect of their proposed relation on a global scale, the corresponding relation (Eq. 4, Fig. 2) is included in the TRSP simulation. Note that number processing is still ignored and characteristics of scavenged aerosols are retained upon release.

As expected (since the resuspended aerosol fraction is now lower than the evaporation fraction), the atmospheric burden for all species is lower in the TRSP simulation than in the SMPL simulation (Table 3). The effect is again strongest for the completely soluble species sulphate and sea salt (-18% vs. SMPL). The effect on dust now stands out with a distinctly lower



**Figure 8.** Annual average, zonal mean concentrations ($\mu g\ m^{-3}$) of M7 aerosol species in the BASE simulation and differences ($\mu g\ m^{-3}$) compared to the BASE simulations compared in the other simulations. Each row represents one of the aerosol spies, from top to bottom: sulphate (SO$_4$), particulate organic matter (POM), black carbon (BC), mineral dust (DU) and sea salt (SS). Each column corresponds to a single simulation, from left to right: BASE, SMPL, TRSP, PRCS and COMB simulations.



decrease (-10% vs. SMPL) than BC and POM (-15% vs. SMPL). The effect of the new relation is so strong that it offsets the increase between the BASE and SMPL simulation. Compared to the BASE simulation, all species now have a 2.7 to 8.5%

lower atmospheric burden (Table 3).

The average evaporation rate of 60% coincides with the maximum discrepancy between evaporated rainwater and released aerosol in the G06 relation (see Fig. 2). This causes a substantial decrease in aerosol resuspension in the TRSP simulation, which is more than halved compared to the SMPL simulation (i.e. about -60% for SO4, BC, POM, -45% for dust and -70% for seasalt). Deviations between species are caused by the collocation of their individual distributions and the precipitation patterns, as well as by the differences in scavenging efficiencies. The global amount of aerosol removal by scavenging decreases by about -25%. Because the reduced evaporative release decreases the potential of convection and advection to replenish concentrations aloft, the initial removal by scavenging as conceptualised in Fig. 7 now leads to a dominant decrease in the largest part of the atmosphere (Fig. 8). Only in the lower-tropospheric (sub-)tropics, the initial evaporative aerosol release is sufficiently

strong to counteract the increased scavenging. The only change between the TRSP and SMPL simulations is the re-evaporation of precipitation and subsequent resuspension of aerosol. However, the amount of scavenging in the TRSP simulation also decreases substantially compared to the SMPL simulation. This shows that a substantial part of the scavenged aerosol, has been scavenged and released before. Because a smaller fraction of the aerosol is released upon evaporation of the precipitation in the TRSP simulation it is ultimately removed from the atmosphere and not transported by advection and convection to

regions with precipitation where it can be scavenged again.

### 4.2.3 Number processing

In the PRCS simulation release of aerosol is treated following the observations of M92 that evaporated raindrops only release one aerosol. Instead of tracking the number of scavenged particles and releasing them proportional to the evaporated water fraction, the raindrop number concentration is used to calculate the number of released aerosol. For mass resuspension, we

return to the 1:1 relation with evaporated water fraction.

At first glance (Fig. 8), adding number processing has an effect comparable to the G06 relation for mass resuspension in the TRSP simulation. However, there is a distinction between the smaller size aerosols (sulphate, POM, BC) and coarse size aerosols (sea salt, mineral dust) as visible in Table 3. The smaller aerosols show a stronger decrease (-10 to -11%) than in the TRSP simulation. The coarse mode aerosols show a weaker decrease (-0.57% for sea salt), whereas compared to the

BASE simulation the burden of dust aerosol increases by 2.5%. (Table 3). Nitrate aerosol shows a strong increase of 27%. The magnitude of wet deposition processes (scavenging and evaporation) in the PRCS simulation is almost the same as in the SMPL simulation for sulphate, BC and POM, i.e. differences are smaller than 2%. This is drastically different from the effects in the TRSP simulation where the magnitude of wet deposition (both scavenging and evaporation) halves. Instead, the decrease in aerosol burden in the PRCS simulation is caused by sedimentation. This process was negligible for the species that reside on

the smaller (accumulation) size aerosol in previously discussed simulations. However, in the PRCS simulation it becomes an important removal process as smaller particles are moved to the coarse mode upon resuspension. Dry deposition for sulphate, POM and BC increases by a factor of 6.8, 7.6 and 5.8, respectively. This increase is caused by the changes in aerosol distribution

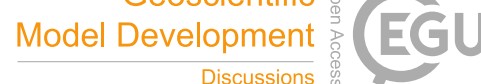



due to the number processing in precipitation. Due to the scale gap in number concentrations of aerosols ($10^4$-$10^6$ m$^{-3}$) and raindrops ($10^3$ m$^{-3}$), each hydrometeor holds a large number of aerosol particles which combine into one large aerosol upon
release. The strength of this process and the increase in aerosol size is reflected in the amount of released particles to the accumulation and coarse modes. Even though all particles are now assigned to these modes, the amount of particles released into these modes decreases by 99.98% and 92.6% respectively. So, the increased removal of aerosol in the PRCS simulation consists of two steps: (1) Increased precipitation causes a stronger downward transport and (2) the aerosols released from the precipitation are considerably larger and are removed from the atmosphere by sedimentation more quickly. This process also occurs for mineral dust (+10.9%) and sea salt (+3.52%), but for these large size aerosol species sedimentation already was an important removal process and the balance between the different removal mechanisms is not altered substantially.

Because nitrate is treated as bulk aerosol in TM5, number processing does not apply to this aerosol species. The process
of wet deposition for this species is therefore similar to the SMPL simulation. However, the less effective wet removal is no longer counteracted by a decrease on chemical production due to changes in sulphate. In fact, the decreased concentration of sulphate allows for more production of ammonium nitrate.

The effects of the G06 relation and number processing are combined in the COMB simulation, making the wet deposition
scheme the physically most realistic representation of all simulations. However, the atmospheric burden of all aerosol species decreases even further compared to the TRSP and PRCS simulations. The combined impact remains strongest for the smaller size aerosols of sulphate, POM and black carbon with a decrease of -16 to -19%. Coarse size aerosols of mineral dust and sea salt show a slightly lower decrease of -11%. Nitrate shows an increase of 7.9% which is substantially lower than the PRCS simulation, because of the more efficient removal when applying the G06 relation. The evaporation near the surface in the
tropics is no longer visible except for mineral dust (Fig. 8).

### 4.3 Impact on AOD

To put the differences between the different model simulation into perspective, the model output is compared to the MODIS retrievals of column integrated AOD and CALIOP retrievals of vertical profiles of aerosol extinction. Fig. 9 shows the annual mean spatial distribution of column total AOD as simulated in the BASE and SMPL simulations and observed by MODIS. The regions with highest AOD, in observations as well as all model simulations, are located in a band running from East China, via India, to the Sahara desert with an outflow to the west over the Central Atlantic Ocean. Lowest AOD in the observations is found in Australia, the southern tips of Africa and South America and drier regions of the United States. In the simulations,
low AOD is found over the ocean, especially near Antarctica.

Comparing AOD in simulations and observations quantitatively, we can conclude that AOD is on average underestimated in the model, i.e. an observed global mean of 0.166 vs. a simulated global mean of 0.107 in the BASE simulation and ranging from 0.084 to 0.125 in the other simulations (Table 3). Here, the (monthly mean) model values are only sampled for gridcells with a valid MODIS AOD in the same gridcell for that month. Because MODIS makes use of reflected solar radiation this



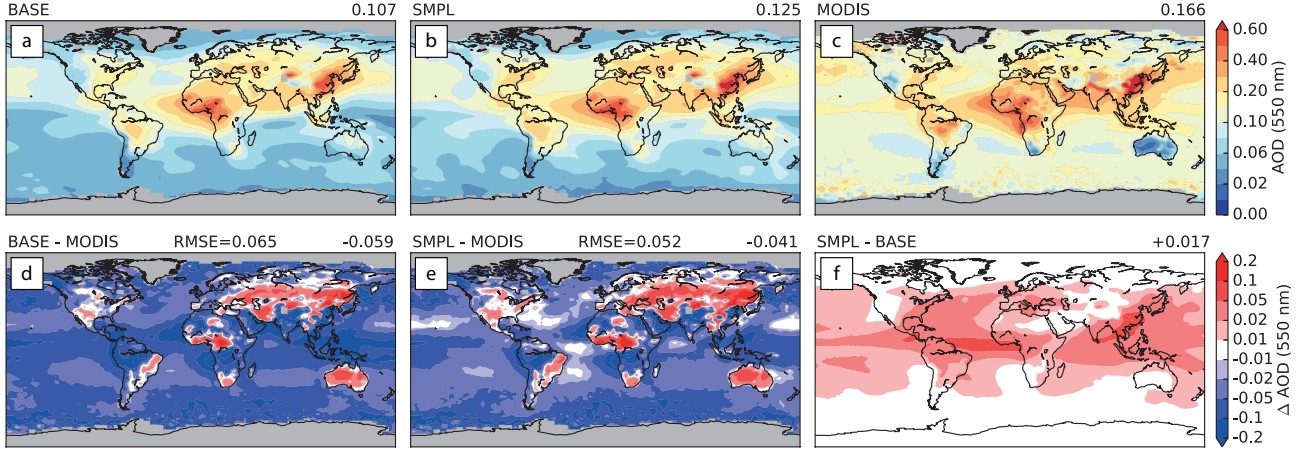

**Figure 9.** Mean column-average AOD in a) the BASE simulation, b) SMPL simulation and c) observed by MODIS in the period March 2005 - Feb 2006. The resulting difference in AOD between d) BASE and MODIS e) SMPL and MODIS and f) SMPL and BASE. The numbers above the top right corners are area-weighted global mean AOD. Model values are sampled only for months and locations with valid MODIS measurements.

means that values are biased towards summer values, since there is insufficient reflected radiation over the winter pole for reliable retrievals.

Pattern-wise, AOD over oceans is generally underestimated, while local overestimation of AOD is only found over land. These regions include the largest part of Asia (except the densely populated areas of East China), South East Asia and India. Additional regions are Australia where observed AOD is exceptionally low and a part of West Africa, which is affected by a

monsoon and consequently influenced by either mineral dust from the Sahara, sea salt from the Atlantic or aerosol resulting from biomass burning at different times of the year.

In the SMPL simulation the atmospheric aerosol burden increases, especially over the oceans in the tropics (related to the high evaporation fraction and higher AOD values), but the gap between model and observation is not closed and AOD is underestimated. General patterns of difference with observations do not change substantially because the changes between

simulations are spatially more homogenous and generally smaller than the mismatch with measurements. The improvement of RMSE, 0.065 to 0.052, is almost completely caused by the decrease of underestimation of AOD over the oceans. All other simulations have a decreased AOD with respect to the BASE simulation, and adjust the AOD in the wrong direction. Also, patterns in the other simulations (not shown) are comparable to the ones shown in Fig. 9.

Notable, is the decrease in global mean AOD (see Table 3) for the simulations including number processing. For the SMPL

and TRSP simulations AOD shows the same decrease in AOD as in aerosol burdens. In contrast, the decrease in AOD in the PRCS and COMB simulations is stronger than the decrease in aerosol burdens. Without number processing the characteristics of the aerosols remains unchanged after resuspension and changes in AOD are therefore proportional to the changes in aerosol burdens. When number processing is applied, the released aerosols are larger as explained in Sect. 4.2.3 and the total aerosol





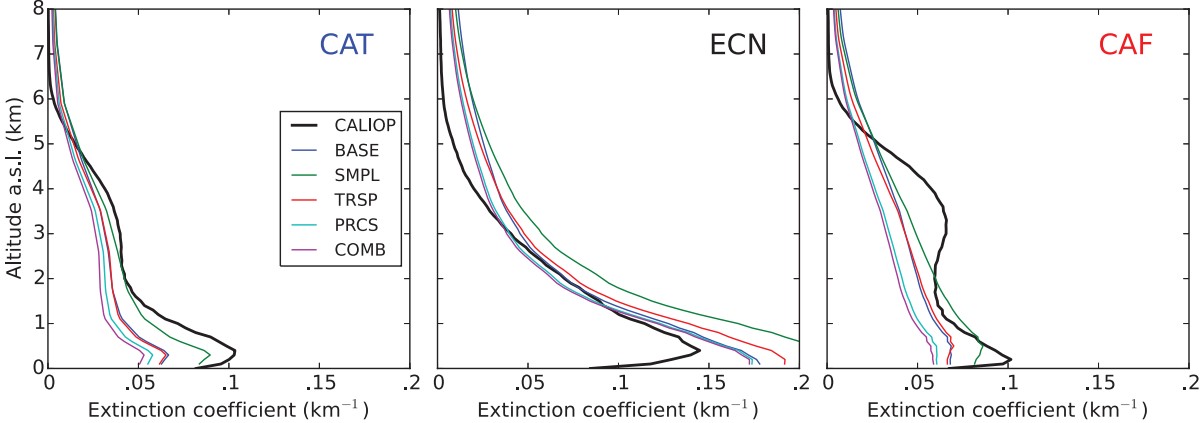

**Figure 10.** Summer (JJA) average extinction coefficient (km$^{-1}$) profiles for 2005 (models) and 2007-2009 for CALIOP observations in the Central Atlantic (CAT), Eastern China (ECN) and Central Africa (CAF) regions as used in Koffi et al. (2012).

mass in the atmosphere contains a lower aerosol number. This has an impact on the AOD evaluated at 550 nm because small

aerosols are more effective in scattering incoming solar radiation.

Comparison of model simulations with AeroCom-CALIOP aerosol extinction profile data from Koffi et al. (2012) provides more insight into model errors in AOD and might indicate what causes the differences. This final comparison will focus on three regions which are also discussed above: the Central Atlantic Ocean (CAT) which is dominated by sea salt aerosol, East China (ECN) which is mainly affected by anthropogenic aerosol, and Central Africa (CAF) which has mostly biomass burning

and desert related aerosol. Shown in Fig. 10 are the vertical aerosol extinction profiles for boreal summer (JJA) in these regions observed by CALIOP and modelled in the different simulations.

Model performance differs depending on the region, but shape and relative magnitudes are captured well. The most prominent and important difference between model and observations in all regions is the overestimation of extinction aloft (> 5 km a.s.l.) even when surface extinction is too low. The introduction of precipitation evaporation and other changes to the repre-

5  sentation of wet deposition does decrease the extinction aloft, but the differences are minor compared to the mismatch with observations at these heights.

In the CAT region, extinction is caused by two aerosol species. Being an oceanic region, sea salt aerosol dominates near the surface and although underestimated in absolute value the vertical positioning of the maximum is captured well by the model. The range between 2 and 5 kilometers a.s.l. is influenced by outflow of aerosol from the African continent and consist of mineral

10  dust. Again, the vertical position in the model is reasonably well reproduced, but the elevation in extinction is underestimated. In the ECN region, extinction is overestimated in the complete vertical column, in accordance with comparison to MODIS observations. The exponential decrease in this region suggest strong local aerosol sources at the surface and and only minor influence by other regions, which would be visible as layers with elevated extinction at higher altitudes. Differences in the CAT





and ECN regions are a general under and overestimation of the complete profile, pointing to a wrong balance between local emission and removal in these regions, but no major errors in underlying patterns or mechanisms.

In the CAF region however, the model misses the aerosol layers between 2 and 5 km completely and displays a linear decrease instead. This indicates difficulties of the model treating the transport of lofted desert dust. Either the model misses the source of this elevated layer or removes the aerosols at this altitude too quickly.

In summary, the representation of aerosol re-evaporation is physically more realistic, but it does not substantially change or improve global AOD nor the vertical profiles or column integrated horizontal patterns of modelled aerosol extinction.

## 5  Discussion and conclusions

The goal of this work was to improve the treatment of aerosol wet deposition in the EC-Earth model and explore the impact of various ways to treat resuspension on the atmospheric aerosol burden. For this, the representation of the wet deposition was improved. We have included additional meteorological fields from the dynamical core IFS to drive the atmospheric chemistry and aerosol model TM5. This lead to a more realistic representation of wet deposition in TM5, and allows for the inclusion of precipitation evaporation and aerosol resuspension. Several sensitivity simulations were performed to disentangle the impact of changes in precipitation, precipitation formation and evaporation and subsequent mass resuspension (based on G06) and aerosol number processing (based on M92) on the amount and distribution of various aerosol components. The main findings can be summarised as follows:

- Including re-evaporation of large-scale precipitation results in an initial increase of 8 to 15% of simulated M7 aerosol burdens depending on the aerosol species.

- The induced downward transport by enhanced scavenging aloft and evaporation below clouds results in substantial redistribution and higher aerosol burdens at low and mid-latitudes.

- Using the relation of Gong et al. (2006) limits the release of aerosol from evaporating precipitation and offsets the initial increase in aerosol burden leading to a decrease of 2.7 to 8.5% in global aerosol burden.

- Assuming release of a single aerosol from each evaporated hydrometeor as observed by Mitra et al. (1992), leads to a transformation of scavenged small-scale aerosol to coarse particles, which enhances removal by sedimentation and hence leads to a lower aerosol burden of small-size $SO_4$, POM and BC aerosol by 10 to 11% (16 to 19% combined with the relation of Gong et al. (2006)).

Like in most global models, wet scavenging of aerosols in TM5 is modelled using a highly parameterised representation and with it, many uncertainties remain. Yet, the design of the different simulations covered (almost) the complete spectrum of the influence of precipitation evaporation. At one end precipitation was merely a transport medium while on the other end aerosols suspended in hydrometeors are completely clumped together and transformed to a soluble state. With this, we established that the effect of precipitation evaporation can vary substantially, i.e. the global aerosol burden varies up to 30% (Table 3).





To further constrain the effect of precipitation evaporation on the aerosol burdens, measurements are necessary that are specifically geared towards understanding the change in characteristics of aerosols between the time of scavenging and resuspension, i.e. differences in aerosol size, number and chemical composition before and after suspension in hydrometeors. Also, calculations done with models using resolutions that allow for resolving clouds, e.g. Large Eddy Simulations (LES) or parcel models, could provide insight in aerosol processing in clouds and precipitation.

Besides tackling general uncertainties persistent in aerosol modelling, several possibilities for improvement of the representation of aerosol processes remain within the EC-Earth model.

Before discussing possible improvements, it must be noted that the model configuration in this study has not been fully tuned regarding the representation of aerosols. The current set of parameters is based on the standalone TM5 model driven by meteorological fields of the ERA-Interim reanalysis. This set may not be optimal for use in free or nudged EC-Earth simulations. Without considering the possibilities for re-tuning, it is difficult to judge whether the simulations have improved or not. For example, emissions of mineral dust are parameterised using surface winds and known to be very sensitive to changes in the local strength of these winds. However, these are quite different in IFS than in ERA-Interim, even if nudging is applied. Another important parameter is the mixing timescale used to account for the incomplete coverage of grid cells by large-scale clouds and precipitation. This parameter determines the strength of (sub-grid) mixing between the cloudy and cloud-free part of a grid cell and changing this parameters will directly change aerosol concentrations and AOD.

Concerning scavenging, a separate treatment of aerosol mass and number for in-cloud scavenging (Croft et al., 2010) could be beneficial. Larger aerosols are activated (and thus scavenged) at a higher rate and the difference in scavenging between the lower and upper bound of an aerosol mode can be substantial. Even though M7 uses a modal description of aerosol, this effect can be simulated by assigning different scavenging rates to mass and number. When aerosol mass is scavenged more effectively than aerosol number, their ratio in the remaining distribution will change to contain relatively more small-scale aerosol that are more prone to remain in the atmosphere. This procedure is already applied to below-cloud scavenging, but not yet for in-cloud processes.

Additionally, in this work only large-scale precipitation was investigated and the most important process negating the effect of evaporative release in large-scale wet deposition was scavenging by convective precipitation (Appendix B). As shown, aerosol optical depth in TM5 is too low in most regions. Including precipitation evaporation in the representation of convective scavenging might reduce AOD biases in these regions by providing a pathway for aerosols to remain in the atmosphere.

An alternative approach to improve wet deposition could include an integral treatment of precipitation formation and scavenging. Currently the calculation of precipitation formation is separated from aerosol scavenging, whereas in reality the interaction of condensating water on aerosol particles creates clouds. Combining these processes allows for tracking of aerosols in clouds and precipitation and enables the calculation of the characteristics of these aerosols when released from a hydrometeor. A new version of the EC-Earth model is currently under development that will use the aerosol number and mass concentrations from TM5 to calculate cloud activation in IFS. This representation will be used for the Aerosol Chemistry Model Intercomparison Project (AerChemMIP, Collins et al. (2017)) as part of the WCRP Coupled Model Intercomparison Project Phase 6

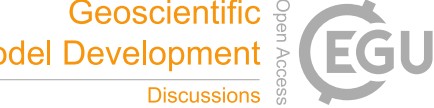

(CMIP6, Eyring et al. (2016)). Although computationally demanding, the inclusion of spectral (bin) microphysics might be necessary to achieve a level of detail necessary to determine the impact of clouds on the characteristics of aerosol distributions.

30  Bulk or modal microphysics schemes might not be equipped to properly simulate the details of the aerosol-cloud interaction (e.g. Khain et al., 2015; Glassmeier et al., 2017).

Improvements in the description must also be found in processes other than wet deposition. The deviations of modeled AOD showed substantially higher variability than the patterns of change between the different representations of wet deposition. This shows that a part of the remaining inconsistencies in the modelling of aerosols with EC-Earth is likely not due to deficiencies in the representation of wet deposition. The general underestimation over the oceans can have multiple causes. Either the model removes aerosol from the atmosphere too quickly in general, or there is a discrepancy in the emission of aerosol from the oceans. One possibility is that the emission of sea salt is too low or the simulated size distribution creates too large

particles, which are then too efficiently removed from the atmosphere by sedimentation or wet deposition. Another cause could be an underestimation of dimethyl sulfide (DMS) emissions from the oceans which directly translates into a lower sulphate concentration.

The regions of over- and underestimation seems to be attributable to specific features, (e.g. Asian boreal forest or India) and the strength of local sources are worth investigating. Also, the incapability of the model to simulate lofted aerosol layers in the

CAT and CAF regions extends beyond the influence of wet deposition. Together with the overestimation of extinction at high altitudes this points to a too fast mixing of aerosol in the vertical. This might be the effect of the convection scheme in TM5 and needs further investigation.

## 6  Code and data availability

The usage of and access to the EC-Earth source code is licensed to affiliates of institutions that are members of the EC-

Earth consortium. More information can be found at http://www.ec-earth.org. The model version used in this paper can be found under the branch r2875-wet-dep-tm5, which is part of the main development version of EC-Earth updated until 9 September 2016, altered by the changes described in this paper. Observational datasets can be downloaded from respective websites. CALIOP Aerosol Extinction Profile data: http://aerocom.met.no/databenchmarks and MODIS Atmosphere products: https://modis-atmosphere.gsfc.nasa.gov/

**Appendix A:  Derivation of the number of evaporated particles using Marshall-Palmer raindrop size distribution**

We use the Marshall-Palmer raindrop size distribution (RSD) as defined in Marshall and Palmer (1948), which is described as:

$$n_n(D) = N_0 \exp(-\lambda D) \tag{A1}$$





with slope parameter $\lambda = 41 R^{-0.21}$ mm$^{-1}$ and droplet number concentration per unit diameter $N_0 = 8 \times 10^4$ mm$^{-1}$ m$^{-3}$. Using this expression, the total number concentration can be calculated as:

$$N_T = \int\limits_0^\infty n_n(D)\,\mathrm{d}D = \int\limits_0^\infty N_0 \exp(-\lambda D)\,\mathrm{d}D = \frac{N_0}{\lambda} \tag{A2}$$

Similarly, the total mass concentration is given by:

$$M_T = \int\limits_0^\infty \frac{\pi}{6} \rho D^3 n_n(D)\,\mathrm{d}D = \frac{\pi \rho N_0}{6} \int\limits_0^\infty D^3 \exp(-\lambda D)\,\mathrm{d}D = \frac{\pi \rho N_0}{\lambda^4} \tag{A3}$$

The goal is to calculate the amount of evaporated droplets $N_{evap}$ for a given evaporated fraction $f$ of the total precipitation (mass). The relation between $f$ and $N_{evap}$ is dictated by the assumption of where the water mass is lost. One possible assumption is that the evaporated mass scales with the droplet surface, or equivalently: all droplets have the same decrease in radius

$D_c$. Thus, all droplets with a diameter $\leq D_c$ will completely evaporate and the new total mass can be calculated as:

$$M_{new} = \frac{\pi \rho N_0}{6} \int\limits_{D_c}^\infty (D - D_c)^3 \exp(-\lambda D)\,\mathrm{d}D \tag{A4}$$

$$= \frac{\pi \rho N_0}{\lambda^4} \exp(-\lambda D_c) \tag{A5}$$

Thus the evaporating fraction $f$ can be expressed as a function of $D_c$ as follows:

$$M_{new} = (1 - f) M_T \tag{A6}$$

$$\frac{\pi \rho N_0}{\lambda^4} \exp(-\lambda D_c) = (1 - f) \frac{\pi \rho N_0}{\lambda^4} \tag{A7}$$

$$D_c = \frac{\log(\frac{1}{1-f})}{\lambda} \tag{A8}$$

The number concentration of droplets with an original size smaller than $D_c$ that will evaporate is given by:

$$N_{evap} = \int\limits_0^{D_c} N_0 \exp(-\lambda D)\,\mathrm{d}D = \frac{N_0}{\lambda}(1 - \exp(-\lambda D_c)) \tag{A9}$$

Substituting $D_c$ from Eq. (A8) gives the final link between $N_{evap}$ and $f$:





$$N_{evap} = \frac{N_0}{\lambda} \left( 1 - \exp(-\lambda \frac{\log(\frac{1}{1-f})}{\lambda}) \right) \tag{A10}$$

$$= f\frac{N_0}{\lambda} \tag{A11}$$

So despite the non-trivial intermediate steps, the final result for number concentration of evaporated droplets is given by a simple expression

$$N_{evap} = f\frac{N_0}{\lambda} = fN_T \tag{A12}$$

**Appendix B: Overview of and zonal patterns of processes influencing tracer concentration**

An overview of removal and source processes (scavenging, evaporation, advection, convection, sedimentation, chemical production and emission) for each aerosol species in each simulation is given in Table B1. These global averages are referred to in the main text.

The output of TM5 includes zonal averages of the fluxes associated with the mentioned processes for all model levels. This output provides deeper insight into the complex interaction between processes. To illustrate this, Fig. B1 shows the patterns of

individual atmospheric processes influencing the concentration of particulate organic matter. Despite a different atmospheric distribution of other species, the general patterns and responses of the individual processes are similar to the one shown in this example. Also, of the 5 different simulations, only the BASE, SMPL and PRCS simulations are shown. Patterns in the TRSP (COMB) simulation are comparable to the SMPL (PRCS) simulation despite the weaker evaporation as a result of the G06 relation. Positive (negative) values indicate a local source (sink) of a tracer resulting from an individual process. The

bottom panels show the difference in these patterns in the additional simulation compared to the BASE simulation. Here, positive (negative) values indicate a weaker (stronger) local removal or stronger (weaker) local source. These changes in patterns represent the feedbacks of the different atmospheric processes on the changes made in the representation of aerosol wet deposition.

As discussed in the main text, the amount of scavenging increases because the new representation no longer rescales precip-

itation, which was necessary in the BASE simulation to compensate for neglecting evaporation (panels b1 and c1). At the same time, evaporation is introduced (b2 and c2), leading to a net downward flux of aerosol (b3 and c3). The difference with the BASE simulation is strongly anti-correlated to the change in wet deposition. Generally, advection transports tracers upward, away from the surface and deposits these aloft. This motion becomes stronger and partly removes the surplus near the surface and replenishes the regions where scavenging removed additional tracer material. A similar response is seen for convection,

which also transports tracer material from the surface to higher altitudes. However, this motion is stronger and lifts the material to altitudes above 400 hPa. Sedimentation is almost negligible in the BASE simulation as well as the SMPL where no aerosol processing takes occurs. In the PRCS simulation, however, removal by sedimentation is a substantial sink (f3). The restriction of only releasing a single aerosol particle from an evaporated hydrometeor implies that all suspended aerosols clump together.





**Figure B1.** Zonal average distribution of local tendency in particulate organic matter (POM) in ppb year$^{-1}$ caused by large-scale scavenging, evaporation, wet deposition (scavenging+evaporation), advection, convection and sedimentation. General patterns are shown for the BASE simulation (rows a,d) where positive (negative) values indicate a local source (sink) by the respective proces. Differences compared to these BASE patterns are shown for the SMPL (rows b, e) and PRCS (rows c,f) simulations. Positive (negative) differences indicate a weaker (stronger) local removal or stronger (weaker) local source.



**Table B1.** Overview of global total annual change (Tg yr$^{-1}$) due to individual (removal) processes for all aerosol species. Positive numbers indicate a source, whereas negative number indicate a removal process. The change with respect to the BASE simulation is denoted in italic font

| | BASE | SMPL | Δ | TRSP | Δ | PRCS | Δ | COMB | Δ |
|---|---|---|---|---|---|---|---|---|---|
| **Sulphate (SO$_4$)** | | | | | | | | | |
| LSP Scavenging | -79.64 | -165.4 | *-85.8* | -123.3 | *-43.63* | -162.9 | *-83.21* | -126 | *-46.39* |
| LSP Evaporation | 0 | +97.65 | | +36.65 | | +98.22 | | +40.45 | |
| Convective scavenging | -49.59 | -71.27 | *-21.68* | -53.47 | *-3.876* | -53.55 | *-3.964* | -46.75 | *+2.839* |
| Sedimentation | -3.275 | -4.014 | *-0.738* | -3.252 | *+0.023* | -25.42 | *-22.14* | -11.13 | *-7.858* |
| Production | +127.7 | +138.3 | *+10.57* | +138.6 | *+10.83* | +138.8 | *+11.06* | +138.6 | *+10.92* |
| Emission | +4.711 | | | | | | | | |
| | | | | | | | | | |
| **Particulate organic matter (POM)** | | | | | | | | | |
| LSP Scavenging | -50.56 | -107.9 | *-57.30* | -80.95 | *-30.38* | -108.3 | *-57.72* | -83.16 | *-32.60* |
| LSP Evaporation | 0 | +69.70 | | +30.14 | | +71.44 | | +33.10 | |
| Convective scavenging | -46.65 | -58.80 | *-12.15* | -46.48 | *+0.163* | -46.56 | *+0.092* | -41.03 | *+5.615* |
| Sedimentation | -1.82 | -2.076 | *-0.256* | -1.756 | *+0.063* | -15.66 | *-13.84* | -7.981 | *-6.161* |
| Emission | +98.80 | | | | | | | | |
| | | | | | | | | | |
| **Black carbon (BC)** | | | | | | | | | |
| LSP Scavenging | -4.718 | -9.737 | *-5.019* | -7.262 | *-2.544* | -9.697 | *-4.979* | -7.427 | *-2.709* |
| LSP Evaporation | 0 | +6.003 | | +2.480 | | +6.102 | | +2.723 | |
| Convective scavenging | -3.356 | -4.316 | *-0.960* | -3.305 | *+0.051* | -3.357 | *-0.001* | -2.881 | *+0.475* |
| Sedimentation | -0.192 | -0.217 | *-0.024* | -0.182 | *+0.010* | -1.318 | *-1.126* | -0.687 | *-0.495* |
| Emission | +8.253 | | | | | | | | |
| | | | | | | | | | |
| **Mineral dust (DU)** | | | | | | | | | |
| LSP Scavenging | -278.8 | -739.9 | *-461.1* | -546.0 | *-267.2* | -708.9 | *-430.1* | -497.3 | *-218.5* |
| LSP Evaporation | 0 | +525.4 | | +281.8 | | +503.6 | | +254.3 | |
| Convective scavenging | -65.74 | -103.4 | *-37.64* | -77.45 | *-11.71* | -96.30 | *-30.56* | -79.72 | *-13.97* |
| Sedimentation | -389.3 | -415.8 | *-26.49* | -392.2 | *-2.862* | -432.0 | *-42.68* | -411.2 | *-21.85* |
| Emission | +728.0 | | | | | | | | |
| | | | | | | | | | |
| **Sea salt (SS)** | | | | | | | | | |
| LSP Scavenging | -892.9 | -1425 | *-532.2* | -1138 | *-245.3* | -1265 | *-372.4* | -1099 | *-205.9* |
| LSP Evaporation | 0 | +632.1 | | +186 | | +545.2 | | +173.5 | |
| Convective scavenging | -300.1 | -359.3 | *-59.19* | -293.5 | *+6.657* | -322.7 | *-22.59* | -283.3 | *+16.85* |
| Sedimentation | -4261 | -4302 | *-40.75* | -4208 | *+52.72* | -4411 | *-150.2* | -4245 | *+15.72* |
| Emission | +5454 | | | | | | | | |
| | | | | | | | | | |
| **Nitrate (NO$_3$)** | | | | | | | | | |
| LSP Scavenging | -2.159 | -3.785 | *-1.626* | -3.593 | *-1.434* | -5.172 | *-3.013* | -4.039 | *-1.880* |
| LSP Evaporation | 0 | +1.566 | | +0.756 | | +2.423 | | +0.974 | |
| Convective scavenging | -0.912 | -1.005 | *-0.092* | -0.935 | *-0.023* | -1.146 | *-0.234* | -0.985 | *-0.073* |
| Sedimentation | -2.220 | -2.141 | *+0.079* | -2.083 | *+0.137* | -2.252 | *-0.032* | -2.108 | *+0.112* |
| Production | +5.267 | +5.343 | *+0.076* | +5.836 | *+0.569* | +6.130 | *+0.863* | +6.142 | *+0.875* |





The resulting aerosols are substantially larger than non-processed POM aerosol, making sedimentation significant for these aerosols. The emergence of this process is reflected in a decrease in the removal by advection and convection near the surface.

Noticeable and illustrative of the negative feedback is the isolated region of increased removal by convection near the

5 equator between 600 and 700 hPa for the SMPL simulation (e2). The same region is visible in the change in wet deposition (b3). Although scavenging increases, more tracer material is released by evaporation. This being a region where convection reduces POM, the surplus is partly taken away by convection and released above at 200 hPa.

*Competing interests.* The authors declare that they have no conflict of interest.

*Acknowledgements.* This work is supported by the Netherlands Organization for Scientific Research (NWO), project number GO/13-01. The

10 computations were carried out on the Dutch national supercomputer Cartesius, and we thank SURFSara (www.surfsara.nl) for their support.





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
