# Peer review of "The impact of precipitation evaporation on the atmospheric aerosol distribution in EC-Earth v3.2.0"

_Geoscientific Model Development, 2017_

## Referee Comment (RC1) · Anonymous Referee #1 · 10 Jan 2018

There are still large uncertainties to simulate the aerosol distribution, especially for the aerosol-cloud interaction. Aerosol-cloud interaction significantly affects the aerosol vertical distribution. The impact of precipitation evaporation on the atmospheric aerosol distribution in EC-Earth model is investigated by implementing several more physical and realistic parameterizations in this manuscript. The results are interesting. I recommend to accepting it after minor revision as indicated below.

1. Page 4 Line 27, Aerosol species are assumed as a complete internal mixture in each mode, what do you mean the mass of the included species are tracked by separate prognostic variables? Please clarify it. 2. Page 4 Line 29, How the prognostic total

[Figure]

number of aerosols of each mode is calculated in the TM5? 3. Page 5 Line 1, how do you calculate the optical properties due to the condensed ammonium nitrate? 4. Page 5, Line 21, How do you set the time step for the TM5, 6 hours or not? What do you mean of the next time step and the artificially introducing mixing? 5. Page 6, Line 11, Does the coupler only exchange the meteorological fields at the time snapshot of only every 6 hours? How about the intermediate fields at every 45 minutes? 6. Page 8, Line 17, How does the IFS calculate the evaporated precipitation fraction? 7. How do you compare the simulated AODs with MODIS? Do you consider the time inconsistent? MODIS combined Terra and Aqua generally only have twice observations per days.

———————————————————

---

## Referee Comment (RC2) · Anonymous Referee #2 · 16 Jan 2018

This manuscript "The impact of precipitation evaporation on the atmospheric aerosol distribution in EC-Earth v3.2.0" by de Bruine et al. presents a model study on how aerosol removal by clouds and the subsequent vertical redistribution of aerosol in precipitation affects the simulated 3D global aerosol fields. They evaluate model approaches of different complexity for representing these effects. The topic is interesting and the methods presented in the paper can be useful for global aerosol modellers. The paper is clearly and well written and thus easy to follow. I can recommend publishing the paper after the following minor points, that have been detailed below, have been addressed. The line numbering is a bit confusing and in the following comments I refer to the line number indicated in the margin of the pdf.

- Page 2, Line 23: Change "aerosol distribution" to "aerosol size distribution".

- Page 5, Lines 4-5: This sentence "The large-scale precipitation is described using variables like cloud cover and water content" is very ambigous and it is not clear how this actually is distinct from that of convenctive precipitation. Cloud cover describes a sub-grid scale property of a cloud and I also assume that the large-scale precipitation is parameterised.

- Page 5, Line 14: I assume that these are boundaries categorizing warm, mixed, and ice clouds. However, it could be more clearly stated.

- Page 5, the last line of the page: What do you mean by "partly"?

- A more appropriate location for Section 2.5 would be after the model description (at the end of Section 3).

- Page 11, Equation 10: Why do you calculated the mean density volume weighted as opposed to mass weighting?

- Page 19, Line 7: I don't understand this sentence "This shows that a substantial part of the scavenged aerosol, has been scavenged and released before." Before what?

- Page 12-14: Change "raindrops only release one aerosol" to "each raindrop releases one aerosol".

- Section 4.3: Which MODIS product do you use?

- Page 20, the last line of the page: What do you mean by "a valid MODIS AOD"?

- Page 20: Why don't you collocate all time instances of the model AOD to when there is a MODIS observation (see e.g. Schutgens, N. A. J., Partridge, D. G., and Stier, P.: The importance of temporal collocation for the evaluation

of aerosol models with observations, Atmos. Chem. Phys., 16, 1065-1079, https://doi.org/10.5194/acp-16-1065-2016, 2016. )

- Page 22, Figure 10: Please add the uncertainties of CALIOP observations to the figure.

- Page 22, Line 29: What do you mean by small aerosols? The aerosol particles that are the most efficient scatterers of 550 nm solar radiation are few hundred nanometers in diameter while the smallest particles have a very small radiative effect.

- Page 22, Line 37: what do you mean by "relative magnitudes"?

- Page 23, Line 15: what do you mean by "underlying patterns or mechanisms"? How do you deduce that they don't have any major errors?

- Please also do a thorough language check (proof reading + grammar check)

---

## Author Comment (AC1) · 13 Feb 2018

**Reply to Anonymous Referee #1**

We would like to thank the reviewer for the comments and careful reading of the manuscript. Point-by-point replies to the comments are provided below.

1.  *Page 4 Line 27, Aerosol species are assumed as a complete internal mixture in each mode, what do you mean the mass of the included species are tracked by separate prognostic variables? Please clarify it.*
2.  *Page 4 Line 29, How the prognostic total number of aerosols of each mode is calculated in the TM5?*

The first two questions of the reviewer both concern the representation of aerosols in the M7 scheme. M7 includes five aerosol species (sulphate, black carbon, particulate organic matter, sea salt and mineral dust) distributed over 7 lognormal modes. Each species has multiple prognostic tracers for mass, one for each mode the species exists in. For example, sea salt is assumed to exist in 2 modes: accumulation soluble mode (ACS) and coarse soluble mode (COS); therefore 2 prognostic tracers are used to describe the mass of sea salt aerosol. In total, there are 18 prognostic tracers for aerosol mass. An overview of the distribution of the species over the modes is shown in Figure 3 on page 12 of the manuscript. Additionally, because the aerosol modes are assumed to be internally mixed, each mode has one prognostic variable for aerosol number, so 7 in total. Altogether, M7 uses 25 prognostic tracers to describe atmospheric aerosol. More details can be found in the original paper describing the M7 scheme (Vignati et al., 2004). We adapted the corresponding line in the manuscript to clarify the prognostic variables used in M7 and refer to Figure 3 in the text.

3.  *Page 5 Line 1, how do you calculate the optical properties due to the condensed ammonium nitrate?*

Ammonium nitrate is not described by the M7 scheme. A separate routine called EQSAM is used to calculate the partitioning of semivolatile species like ammonium nitrate. The mass of condensed ammonium nitrate is added to the M7 ACS mode. For the calculation of aerosol optical properties, ammonium nitrate is treated similar to sulphate, i.e. the same refractive index is used. This value is taken from OPAC (Hess et al., 1998) and is based on a solution of 75% sulphuric acid. This way, ammonium nitrate has the same impact on the radiative properties as sulphate per unit volume. We have clarified the text and refer to the paper describing the routine: Aan de Brugh et al. (2011).

4.  *Page 5, Line 21, How do you set the time step for the TM5, 6 hours or not? What do you mean of the next time step and the artificially introducing mixing?*

The maximum duration of a time step in TM5 is set to 1 hour. Additionally, the Courant-Friedrichs-Lewy(CFL) criterion is applied and time steps are shortened when the threshold is exceeded. The meteorology that TM5 uses, however, is only updated every 6 hours. Consequently, cloud cover, precipitation and thus scavenging strength do not change at every time step in TM5.

In TM5, the removal due to scavenging by large-scale clouds and precipitation is reduced in proportion to the cloudy fraction of a gridbox. However, since the model doesn't have separate tracers for the cloudy and clear parts of the gridbox, this removal reduces the gridbox total amounts. In the subsequent time step the aerosol concentration in the cloudy and clear part is again the same and the aerosols have been 'numerically' moved/mixed from the clear part of the gridbox to the cloudy part. As a consequence, scavenging would remove aerosol too fast from partly clouded grid boxes. To compensate for this, a mixing time scale is introduced, which effectively delays the

mixing between the clear and cloudy parts within the gridbox (see e.g. Vignati et al., 2010). This mixing time scale is set to 6 hours.

The description of TM5 (Section 2.2) has been adjusted to explicitly mention the time stepping of TM5.

5. *Page 6, Line 11, Does the coupler only exchange the meteorological fields at the time snapshot of only every 6 hours? How about the intermediate fields at every 45 minutes?*

Coupled meteorological fields are a time average of the preceding 6 hours prior to the time of exchange. We have clarified the description of the timing of the meteorological fields in Section 2.4.

6. *Page 8, Line 17, How does the IFS calculate the evaporated precipitation fraction?*

IFS does not explicitly calculate evaporated precipitation fraction. Instead, this quantity is diagnosed in the wet deposition routine of TM5 using the values of falling precipitation and precipitation evaporation.

Text has been adjusted to explicitly state that evaporated precipitation fraction is calculated in TM5.

7. *How do you compare the simulated AODs with MODIS? Do you consider the time inconsistent? MODIS combined Terra and Aqua generally only have twice observations per days.*

In this work, simulated AOD is compared to MODIS observations on a monthly mean basis. We do agree with the reviewer that collocation of model results with MODIS overpasses makes the evaluation more consistent. However, such a detailed analysis requires hourly output, slowing down the model considerably. Because the main focus of this work is to address the importance of including the effects of precipitation evaporation and introducing a method to implement this process in a global modal, the choice was made to produce monthly means from daily mean AOD to be compared to the monthly mean MODIS product. For a fair comparison that justifies complete collocation of the simulations and observations, other uncertainties of the aerosol emissions should also have to be addressed and the model would have to be re-tuned.

References

Aan de Brugh, J. M. J., Schaap, M., Vignati, E., Dentener, F., Kahnert, M., Sofiev, M., Huijnen, V., and Krol, M. C.: The European aerosol budget in 2006, Atmos. Chem. Phys., 11, 1117–1139, doi:10.5194/acp-11-1117-2011, 2011.

Hess, M., Koepke, P. and Schult, I. (1998) Optical Properties of Aerosols and Clouds: The Software Package OPAC. Bulletin of the American Meteorological Society, 79, 831-844.

Vignati, E., Wilson J., and Stier P.: M7: An efficient size-resolved aerosol microphysics module for large-scale aerosol transport models, J. Geophys. Res., 109, D22202, doi:10.1029/2003JD004485, 2004

Vignati, E., Karl, M., Krol, M., Wilson, J., Stier, P., and Cavalli, F.: Sources of uncertainties in modelling black carbon at the global scale, Atmos. Chem. Phys., 10, 2595-2611, doi:10.5194/acp-10-2595-2010, 2010

---

## Author Comment (AC2) · 13 Feb 2018

**Reply to Anonymous Referee #2**

We thank the reviewer for the comments and careful reading of the manuscript. We address the minor concerns in detail below.

*1.  Page 2, Line 23: Change "aerosol distribution" to "aerosol size distribution".*

The text has been adjusted.

*2.  Page 5, Lines 4-5: This sentence "The large-scale precipitation is described using variables like cloud cover and water content" is very ambiguous and it is not clear how this actually is distinct from that of convective precipitation. Cloud cover describes a sub-grid scale property of a cloud and I also assume that the large- scale precipitation is parameterised.*

LSP used for the wet deposition routine is derived from IFS variables coupled to TM5, being liquid/ice water content and cloud cover. In the new deposition scheme, this is extended by variables of falling liquid/ice precipitation, together with ice/liquid formation and evaporation.

However, aerosol transport and scavenging by convective precipitation uses a different approach and is based on entrainment and detrainment fields.

We have clarified the text as follows:

"Scavenging due to large-scale precipitation is derived from prognostic precipitation variables from IFS coupled to TM5, i.e. liquid and ice water content and cloud cover, extended in this work by the variables liquid and ice precipitation, precipitation formation and evaporation. Aerosol transport and scavenging by convective precipitation uses a different approach and is based on entrainment and detrainment variables of IFS."

*3.  Page 5, Line 14: I assume that these are boundaries categorizing warm, mixed, and ice clouds. However, it could be more clearly stated.*

We have clarified the text. It now reads:

"In the current version, in-cloud scavenging is different for liquid, mixed and ice stratiform clouds (Stier et al., 2005). This distinction is based on the local temperature (Croft et al., 2010), where clouds are assumed pure liquid above $0°C$ and pure ice below $-35°C$. Between these boundaries, the clouds are classified as mixed as shown in Table 1."

*4.  Page 5, the last line of the page: What do you mean by "partly"?*

'Partly' refers to the exponentially decreasing factor to scale down the scavenging efficiencies that compensates for the erroneous scavenging of the clear part of the grid box. In hindsight, this only confuses the text as it is introduced afterwards and has been removed. We have adjusted the description (see also reply to Reviewer 1, comment 4).

*5.  A more appropriate location for Section 2.5 would be after the model description (at the end of Section 3).*

Following the advice of the reviewer, the paragraph describing the observational data has been moved to the end of Section 3.

6. *Page 11, Equation 10: Why do you calculate the mean density volume weighted as opposed to mass weighting?*

By definition of mass density being mass over volume, volume weighting has to be applied to correctly calculate the mean density of an internally mixed aerosol.

7. *Page 19, Line 7: I don't understand this sentence "This shows that a substantial part of the scavenged aerosol, has been scavenged and released before." Before what?*

'Before' points to a previous cycle of scavenging and resuspension. The sentence has been removed, as its message is the same as the next.

8. *Page 19, Line 12-14: Change "raindrops only release one aerosol" to "each raindrop re- leases one aerosol".*

The text has been adjusted.

9. *Section 4.3: Which MODIS product do you use?*

In this work, the combined Dark Target and Deep Blue retrievals of MODIS Level 3 monthly mean 1x1 gridded product is used. The text has been adjusted and now refers to the section describing the observational datasets.

10. *Page 20, the last line of the page: What do you mean by "a valid MODIS AOD"?*

Not all grid cells are assigned an AOD value in the MODIS product, i.e. in winter not enough sunlight is available for reliable retrievals. Model data in the affected grid cells are excluded for these instances in the calculation of the annual mean AOD.

The sentence has been changed to:

"Here, the (monthly mean) model values are only sampled for grid cells where MODIS AOD retrievals are available."

11. *Page 20: Why don't you collocate all time instances of the model AOD to when there is a MODIS observation (see e.g. Schutgens, N. A. J., Partridge, D. G., and Stier, P.: The importance of temporal collocation for the evaluation of aerosol models with observations, Atmos. Chem. Phys., 16, 1065-1079, https://doi.org/10.5194/acp-16-1065-2016, 2016.)*

We agree with the reviewer that collocation of model results with MODIS overpasses improves the evaluation. However, such a detailed analysis requires hourly output, slowing down the model considerably. Because the main focus of this work is to address the importance of including the effects of precipitation evaporation and introducing a method to implement this process in a global modal, the choice was made to produce monthly means from daily mean AOD to be compared to the monthly mean MODIS product. For a fair comparison that justifies complete collocation of the simulations and observations, other uncertainties of the aerosol emissions should also have to be addressed and the model would have to be re-tuned.

12. *Page 22, Figure 10: Please add the uncertainties of CALIOP observations to the figure.*

For the comparison with CALIOP retrievals we use the results derived in Koffi et al. (2012). These data are stored as a benchmark on the Aerocom website

). These data are grouped by season and do not include uncertainty estimates. Since we do have data for a number of years, the spread between the years can be used as an indication for uncertainty. The figure has been adjusted to include the maximum and minimum seasonal mean AOD value found in the benchmark data (see Figure below).

[Figure]

Figure 10. Summer season (JJA) mean extinction coefficient (km$^{-1}$) profiles for 2005 (models) and 2007-2009 for CALIOP observations in the Central Atlantic (CAT), Eastern China (ECN) and Central Africa (CAF) regions as used in Koffi et al. (2012). The grey shaded area indicates the spread between minimum and maximum seasonal values in the CALIOP observations.

*13. Page 22, Line 29: What do you mean by small aerosols? The aerosol particles that are the most efficient scatterers of 550 nm solar radiation are few hundred nanometers in diameter while the smallest particles have a very small radiative effect.*

As the reviewer justly points out, the term 'small' here is ambiguous. Small here would refer to any aerosol smaller than the M7 coarse mode. Because of the scale gap in number concentration between aerosols and raindrops, virtually all resuspensded aerosols are returned to this M7 coarse mode. Aerosols of these sizes are on average less effective scatterers than the smaller-size particles they originate from before scavenging. The sentence has been adjusted to:

"This has an impact on the AOD evaluated at 550 nm because the coarse sized aerosols are less effective in scattering incoming solar radiation than the smaller-size particles they originate from."

*14. Page 22, Line 37: what do you mean by "relative magnitudes"?*

'Relative magnitudes' refers to the magnitude of the extinction coefficients in the different regions, which are relatively well reproduced by the model.

The sentence has been adjusted:

"Model performance differs depending on the region, but vertical profile shape and the difference in magnitude of the extinction coefficient between the regions are captured well."

*15. Page 23, Line 15: what do you mean by "underlying patterns or mechanisms"? How do you deduce that they don't have any major errors?*

'Underlying patterns' refers to large-scale wind patterns or (global) distribution of emission regions. The exponentially decreasing pattern in the ECN region really points to

a local source. If the dominant aerosol source would be outside the region, this would show up in a pattern similar to the CAT and CAF region.

The text has been adjusted: 'underlying large-scale meteorological and/or emission patterns, as these would change the shape of the extinction profile.'

**References**

Croft, B., Lohmann, U., Martin, R. V., Stier, P., Wurzler, S., Feichter, J., Hoose, C., Heikkilä, U., van Donkelaar, A., and Ferrachat, S.: Influences of in-cloud aerosol scavenging parameterizations on aerosol concentrations and wet deposition in ECHAM5-HAM, Atmos. Chem. Phys., 10, 1511–1543, https://doi.org/10.5194/acp-10-1511-2010, 2010.

Koffi, B., Schulz, M., Bréon, F.-M., Griesfeller, J., Winker, D., Balkanski, Y., Bauer, S., Berntsen, T., Chin, M., Collins, W. D., Dentener, F., Diehl, T., Easter, R., Ghan, S., Ginoux, P., Gong, S., Horowitz, L. W., Iversen, T., Kirkevåg, A., Koch, D., Krol, M., Myhre, G., Stier, P., and Takemura, T.: Application of the CALIOP layer product to evaluate the vertical distribution of aerosols estimated by global models: 5 AeroCom phase I results, J. Geophys. Res. Atmos., 117, n/a–n/a, https://doi.org/10.1029/2011JD016858, d10201, 2012.

Stier, P., Feichter, J., Kinne, S., Kloster, S., Vignati, E., Wilson, J., Ganzeveld, L., Tegen, I., Werner, M., Balkanski, Y., Schulz, M., Boucher, O., Minikin, A., and Petzold, A.: The aerosol-climate model ECHAM5-HAM, Atmos. Chem. Phys., 5, 1125–1156, https://doi.org/10.5194/acp-5-1125-2005, 2005.